# In Vitro Cell Transformation Assays: A Valuable Approach for Carcinogenic Potentiality Assessment of Nanomaterials

**DOI:** 10.3390/ijms24098219

**Published:** 2023-05-04

**Authors:** Nivedita Chatterjee, Ernesto Alfaro-Moreno

**Affiliations:** NanoSafety Group, International Iberian Nanotechnology Laboratory, 4715-330 Braga, Portugal; ernesto.alfaro@inl.int

**Keywords:** in vitro cell transformation assays (CTAs), nanomaterials (NMs), carcinogenesis, genotoxicity, epigenetic alterations, epithelial–mesenchymal transition (EMT), cancer stem cells (CSCs)

## Abstract

This review explores the application of in vitro cell transformation assays (CTAs) as a screening platform to assess the carcinogenic potential of nanomaterials (NMs) resulting from continuously growing industrial production and use. The widespread application of NMs in various fields has raised concerns about their potential adverse effects, necessitating safety evaluations, particularly in long-term continuous exposure scenarios. CTAs present a realistic screening platform for known and emerging NMs by examining their resemblance to the hallmark of malignancy, including high proliferation rates, loss of contact inhibition, the gain of anchorage-independent growth, cellular invasion, dysregulation of the cell cycle, apoptosis resistance, and ability to form tumors in experimental animals. Through the deliberate transformation of cells via chronic NM exposure, researchers can investigate the tumorigenic properties of NMs and the underlying mechanisms of cancer development. This article examines NM-induced cell transformation studies, focusing on identifying existing knowledge gaps. Specifically, it explores the physicochemical properties of NMs, experimental models, assays, dose and time requirements for cell transformation, and the underlying mechanisms of malignancy. Our review aims to advance understanding in this field and identify areas for further investigation.

## 1. Introduction

Carcinogenesis refers to the disruption of normal cellular function caused by genomic instability, leading to uncontrolled cellular growth and invasion of surrounding tissues, also known as cellular transformation. The acquisition of malignancy, the invasive and destructive potential of cancer, is typically associated with imbalances between oncogenes (genes related to cell division and survival, hence promoting cancer) and tumor suppressor genes (genes related to the control of cellular growth and inductors of cellular death, hence inhibiting cancer), which are often, but not always, associated with genetic mutations or epigenetic alterations resulting from a wide variety of chemical, physical, or biological insults. The process of transforming normal cells into malignant ones is complex and occurs through multiple stages, including initiation, promotion, and progression. Initiation is the first stage of carcinogenesis, characterized by irreversible genetic alterations, such as mutations in key regulatory genes involved in critical cellular pathways. The promotion stage involves the expression of the genome through promoting agents. Finally, the progression stage is irreversible and is marked by chromosomal instability, leading to malignant growth, recruitment of immune cells, invasiveness, and metastasis [1,2,3]. It can take many years for the clinical manifestation of most types of cancer to develop, and even within the same type of cancer, the specific genes that are mutated can vary among individuals. Despite the numerous phenotypes and complexities of the carcinogenesis process, the major hallmarks of neoplastic diseases include sustained proliferative signals, insensitivity to growth suppressors, evasion of programmed cell death (i.e., apoptosis), enabling of replicative immortality, induction and maintenance of angiogenesis, and activation of invasion and metastasis. In addition to these core hallmarks, the deregulation of cellular energy metabolism and avoidance of immune destruction are two emerging hallmarks of cancer [2]. A recent body of data supports the “epigenetic progenitor” hypothesis, which proposes that polyclonal epigenetic disruption of stem/progenitor cells is a common underlying basis of tumorigenesis [4].

The hallmarks of neoplasm refer to the properties of cancer cells rather than the characteristics of the agents that cause cancer. Generally, substances that disrupt the pathways involved in the hallmarks of cancer are likely to be carcinogenic [5,6]. In 2012, the International Agency for Research on Cancer (IARC) identified “10 key characteristics (KCs)” frequently shared among known human carcinogens. These characteristics include the following abilities of an agent: (1) act as an electrophile either directly before or after metabolic activation; (2) be genotoxic; (3) alter DNA repair or cause genomic instability; (4) induce epigenetic alterations; (5) induce oxidative stress; (6) induce chronic inflammation; (7) be immunosuppressive; (8) modulate receptor-mediated effects; (9) cause immortalization; and (10) alter cell proliferation, cell death, or nutrient supply [5,7].

## 2. Carcinogenicity Assessment

The current standard method for assessing carcinogenicity, which is considered the most genuine, involves long-term animal studies, specifically the two-year rodent assay. Nonetheless, the use of rodent bioassay has raised doubts about human risk evaluations because of genetic disparities and false positive outcomes. Additionally, due to financial and temporal constraints, it is unfeasible to apply this extended assay to a large number of existing and upcoming chemicals. Therefore, the need to reduce animal testing and ethical concerns have accelerated the pursuit of alternative in vitro-based techniques [6,8]. It is currently being investigated whether in vitro mechanistic data could serve as an alternative to the two-year rodent bioassay for assessing carcinogenicity [9,10]. Approaches based on computational toxicology are being developed to prioritize chemicals for targeted testing, as well as to identify gene/pathway targets that are relevant to the progression of human cancer [6]. In particular, the previously mentioned “10 KCs” of human carcinogens offer a standardized method for evaluating mechanistic evidence in identifying cancer hazards. The KCs were found to be distinct from the hallmarks of cancer, and interrelationships among them can be used to strengthen the KCs approach and to improve understandings of environmental carcinogenesis. Upon review of the literature, the expert committee in charge provided more precise explanations of each KCs, identified current and emerging assays and in vivo biomarkers that are capable of measuring them, and made recommendations for future assay developments [7,11].

Carcinogenic agents (chemicals; physical insults, such as X-ray and UV light; or complex materials, such as nanoparticles or complex mixtures) can lead to cancer by means of genotoxic or non-genotoxic effects. Non-genotoxic carcinogens are particularly concerning because they work through indirect mechanisms, such as oxidative stress, inflammatory responses, or epigenetic changes. In contrast, genotoxic carcinogens directly damage DNA, making their carcinogenicity more evident [8]. However, if an in vitro genotoxicity test is negative, further testing for carcinogenicity is not mandatory, maintaining the possibility of undetected non-genotoxic carcinogens. Additionally, no OECD-approved screening methods are currently available for identifying non-genotoxic carcinogens [12]. In recognition of this gap, the OECD has established an expert group tasked with developing integrated approaches to testing and assessing non-genotoxic carcinogens, known as IATA strategies [13].

The IARC categorizes substances as proven carcinogens to humans (Group 1), probably carcinogenic to humans (Group 2A), or possibly carcinogenic to humans (Group 2B) (https://monographs.iarc.who.int/list-of-classifications, Volumes 1–133, Last updated: 2023-03-24 09.21am (CEST)). Approximately 10–20% of substances classified by the IARC are potent (probably, possibly, or proven) human non-genotoxic carcinogens [14]. In brief, assessing the carcinogenic potential of chemicals is crucial for chemical safety. However, accurately predicting and realistically assessing the carcinogenicity of thousands of chemicals, including both genotoxic and non-genotoxic substances, as well as nanomaterials, remains a significant challenge.

Even in the case of in vitro platforms, the ultimate determination of whether a substance is carcinogenic is dependent on the phenotypic identification of cancer hallmarks. The cell transformation assay (CTA) is a crucial experimental technique utilized to evaluate whether cells have undergone malignant transformation, and it is considered a vital tool for determining the phenotypic transformation of cells in assessing carcinogenicity [15,16,17].

## 3. Cell Transformation Assays

Cell transformation assays (CTAs) are a set of experimental methods that measure key characteristics and processes of malignant transformations. In vitro cellular transformation experiments have proven to be more cost effective and time efficient compared to in vivo animal transformation assessments. In vivo models are complex and can involve a multitude of potential variables, such as a mixture of different cell lines, individual health conditions, and nutrition, making it difficult to identify the distinct mechanisms behind cellular transformation [16]. In contrast, by selecting suitable cell lines, it is possible to develop in vitro models that replicate a specific aspect or hallmark of carcinogenesis. Some model systems are based on morphological effects, such as soft-agar colony formation and cell invasion/migration, while others provide molecular-level information, such as genetic and epigenetic changes resulting from cellular transformation. With the enforcement of the Seventh Amendment to the EU Cosmetics Directive 76/768/EEC, it has become more critical in recent years to develop suitable in vitro system-based carcinogenicity assays for the regulatory testing of chemicals. In vitro system-based CTAs can provide a faster, cost-effective, and efficient initial screening of the carcinogenic potential of chemicals. However, it is essential to combine data from a range of CTA assays that include some carcinogenic endpoints (key characteristics of transformed cells) when assessing carcinogens [18].

### 3.1. Essential Characteristics of Transformed Cells

Qualitative or quantitative phenotypic endpoints can be observed in cells that have undergone successful transformation, including lack of functional contact inhibition of growth, anchorage-independent growth, enhanced proliferation rate, potential cellular migration and invasion, cell-cycle deregulation, programmed cellular dead (i.e., apoptosis) resistance, and tumor formation ability, in experimental animals [16,18]. In addition to these key features, transformed cells also exhibit alterations in cell membrane structure and functions, such as changes in cell surface glycolipids, glycoproteins, proteoglycans, and mucins, as well as the modulation of cell–cell communication through gap junctions and extracellular matrix components. Transformed cells also can promote angiogenesis, higher mutation rates, chromosomal changes, DNA repair defects, epigenetic alterations, and maintenance of telomere length [16,18].

### 3.2. CTA, Based on the Characteristics of Transformed Cell

For the past 50 years, the ability of cells to grow and form colonies on semi-solid media, such as agarose or methylcellulose, has been considered the benchmark assay for evaluating pre-neoplastic endpoints. This phenomenon is called anchorage-independent growth. In this assay, both control and transformed cells are cultured on soft agar composed of two layers of different agar concentrations, and the growth of colonies is compared after 3–4 weeks. Normal cells are hindered in their growth as the semi-solid medium prevents them from adhering to the plate surface, while transformed cells form colonies in an anchorage-independent manner [16]. This assay is useful for facilitating time-respective study designs to elucidate dynamic molecular and morphological changes that occur during cell transformation. Furthermore, transformed cells can also be assessed, among the most common, for migration and invasion capability, as well as for gap junctional intercellular communication (GJIC), cell cycle, and apoptosis markers as measurable endpoints of carcinogenic phenotype. Pathway-focused carcinogenesis analysis can be performed through gene/protein expression analysis using genomics or proteomics approaches in a mechanism-based cell transformation assay [18].

The ability of transformed cells to induce tumor formation in an appropriate experimental animal is considered the ultimate criterion for evaluating their malignancy. Therefore, cells that are believed to be transformed cannot be defined as “malignant” if they fail to produce tumors in host animals. In most cases, the induction of xenograft tumors is achieved by injecting transformed cells into immunodeficient mice, which prevents transplant rejection [16].

The use of in vitro CTA is not only valuable for screening potential carcinogenic compounds in the food, drug, and cosmetic industries; more significantly, it provides a platform to comprehend the fundamental mechanisms underlying carcinogenesis through induced cellular transformation in vitro [16]. Furthermore, in certain situations and for specific chemicals, such as receptor-mediated carcinogens and non-genotoxic carcinogens, CTA is a better alternative to standard genotoxicity test batteries [8,18]. Despite the immense potential of CTA, there are reservations about its use due to certain limitations, and it is not yet accepted for regulatory purposes. One of the reasons for this is the large variation in CTA methodologies. Another reason is that the frequency and rate of transformation can be influenced by several factors, such as cell types, chemical dose, and metabolic activation. Additionally, there are ongoing debates about the use of immortalized cell lines in in vitro CTA because they differ intrinsically from normal cells due to deliberately acquired mutations and modifications, which may limit the interpretation of true carcinogenic mechanisms [16,19]. Notwithstanding the potential risks associated with immortalized cells promoting malignancy, studying their ability to do so can yield valuable insights into the underlying mechanisms and in vivo manifestations of cancer [16]. In efforts to identify chemical carcinogenicity, various mammalian cell transformation systems have been assessed, including animal and human cell lines.

The OECD guidelines recommend an integrated analysis of three key parameters, namely colony forming efficiency (CFE), type III foci formation, and anchorage-independent growth, as a powerful in vitro tool to investigate malignant transformation induced by both genotoxic and non-genotoxic carcinogens [20].

### 3.3. Animal Cell Line-Based Cell Transformation Assays

The OECD review focused on three primary cell transformation assays as short-term test platforms for identifying carcinogens. These assays included the Syrian hamster embryo (SHE), BALB/c 3T3 cell, and C3H10T1/2 cell assays. The SHE assay was used to detect early-stage transformations identified by morphological changes, such as random cell orientation and cell pile-ups within the colonies. In contrast, BALB/c 3T3 cell and C3H10T1/2 assays were used to evaluate the later stages of transformation and detect the formation of transformed cell foci on a confluent monolayer. The reliability of these three cell lines in detecting genotoxic carcinogens, tumor promoters, and human carcinogens was satisfactory, with a classification rate of 90% for class 1 and 95% for class 2 by the IARC. The OECD has recommended developing test guidelines for CTAs that use SHE cells (at physiological or acidic pH), as well as the BALB/c 3T3 cell line. Although the CTA using the C3H10T1/2 cell line was considered useful in elucidating molecular mechanisms of cell transformation at the genomic and transcriptomic level, a test guideline was not recommended at that time due to limited data on reproducibility [18,20].

A modified, improved, and inter-laboratory validated cell transformation assay using BALB/c 3T3 cells was proposed to evaluate the initiation and promotion activities of chemicals, including both genotoxic and non-genotoxic carcinogens [21]. Moreover, a cell model named Bhas 42, involving the transfection of the gene v-Ha-ras in the BALB/c 3T3 cell, was developed and successfully used to positively evaluate both non-genotoxic and genotoxic carcinogens, significantly reducing the time required for the cell transformation assay (cell foci formation) [22,23].

It is widely acknowledged that animal-based models are imperfect in recapitulating human carcinogenesis as the tumor development process in humans is markedly different from that in rodents. It is well known that normal human cells undergo a greater number of genetic and epigenetic alterations before transforming into a neoplastic state [24]. Therefore, it is challenging to extrapolate inter-species carcinogenicity and interpret the risk of human carcinogenicity from rodent cell models as they lack mechanistic understanding and face inherent limitations in their applicability to humans. Additionally, certain chemicals can be misclassified by rodent cell transformation assays. For example, a previous study on the SHE-based cell transformation assay reported a high rate of false positives and limited the ability to distinguish between rodent and human carcinogens [25,26]. Nevertheless, some studies have suggested that cell transformation assays are useful tools for predicting carcinogens [17]. In this regard, human cell line-based CTAs could partially overcome the limitations of animal-based models and offer a more predictive platform for evaluating the carcinogenic potential of chemicals in humans [18].

### 3.4. Human Cell Line-Based Cell transformation Assays

One of the essential characteristics required for malignant transformation is cell immortality, which makes the frequency of transformation a crucial factor when selecting appropriate cell lines for in vitro CTA. Although most normal non-immortalized human cell lines are unsuitable due to their low transformation frequency, human bronchial epithelial cells (BEAS-2B cells) have been demonstrated as a suitable non-malignant in vitro model for carcinogenesis studies [16]. The BEAS-2B cell line has been extensively used for in vitro transformation assays, particularly as a long-term (chronic) exposure-induced cell transformation model, even with p53 missense mutation (a single base substitution in the codon 47) and polymorphism (variation in the codon 72), which do not cause dysfunction of p53 [27]. The other types of cell lines that have been successfully applied are immortalized normal human prostate epithelial cells (RWPE-1), human bronchial epithelial cell (16-HBE), early-transformed human mammary (MCF10AT1) cells, human keratinocytes (HaCaT) cell line, human embryo lung fibroblast (HELF) cells, L-02 cells normal human liver (L-02) cells, immortalized human normal non-tumorigenic kidney (proximal tubular epithelial) (HK-2) cells, human pleural mesothelial MeT-5A, human small airway epithelial cells (SAECs). The human cell line-based CTA has been extensively studied for metals, followed by cigarette smoke condensate, Benzo[a]pyrene, coal tar pitch extract, and PM_2.5_, among others. Recent reviews comprehensively reviewed the mechanism and processes involved in metal (arsenic, cadmium, chromium, nickel)-induced carcinogenesis [16].

In summary, the in vitro CTA model can evaluate a substance’s ability to induce pre-neoplastic (such as anchorage-independent growth in soft agar) or neoplastic (such as tumorigenesis when transplanted into nude mice) phenotypes, regardless of the underlying mechanism of cell transformation.

## 4. Engineered Nanomaterials (NM) and Their Carcinogenicity Assessments

Nanomaterials, defined as materials with at least one dimension between 1 and 100 nm (as defined by the European Commission (Regulation (EC) No. 1907/2006)), possess unique properties. The exponential growth in the invention, production, and use of engineered nanomaterials (NMs) has resulted in inevitable exposure to humans and the environment. Nanotechnology has transformed various sectors, including consumer products, textiles, food, packaging, cosmetics, electronics, and biomedical applications, such as disease diagnosis, drug delivery, and tissue engineering. Nanomaterials also are used in optics, sensors, energy, automotive, chemical, aerospace, construction, and environmental industries, among the most relevant [28,29,30].

The unique physicochemical characteristics of NMs, including size, shape, chemical composition, surface area, surface energy, surface roughness, and physiochemical stability, give rise to their vast potential in various sectors, such as healthcare, and industrial applications. NMs can be manufactured from a wide range of chemicals and fine-tuned to meet specific needs. However, the numerous variations in physicochemical characteristics and their mode of interaction with biological systems and the environment also determine their fate and potential risks [29,31]. For instance, the recent use of NMs in agriculture as nano-based fertilizers and herbicides can lead to increased release of NMs into the environment and food chain [32]. Occupational exposure during production and disposal [33], as well as through consumer products and biomedical applications for theranostic purposes [28], could also pose potential risks to human health.

The potential carcinogenicity of NMs is still uncertain, especially considering that they comprise a wide variety of materials. Even in cases where their bulk counterparts are deemed safe, there is the need to test the nanomaterials’ counterpart. For example, while titanium-di-oxide (TiO_2_) bulk particles are considered safe, TiO_2_ nanoparticles are classified as possibly carcinogenic to humans (Group 2B) by the IARC in 2010 [34]. Additionally, carbon black and multi-walled carbon nanotubes (MWCNT)-7 are also categorized as Group 2B carcinogens by the IARC [35]. Despite significant advancements in the field of health and nanosafety, uncertainties remain (28). The interactions between NMs and biological systems have revealed several IARC-proposed KCs [7], such as genotoxicity, DNA repair alteration, oxidative stress induction, immunosuppression, modulation of receptor-mediated effects, and epigenetic marker alteration. A recent publication reported an adverse-outcome pathway (AOP)-anchored assessment of carcinogenicity, specifically for lung carcinogenicity caused by nanoparticles [36], which has documented several endpoints related to IARC-based KCs, such as oxidative stress and inflammation, DNA damage, and mutagenicity, including secondary genotoxicity.

## 5. Application of CTA in the Carcinogenic Potentiality Assessment of Nanomaterials

Increasing efforts to apply CTA when assessing the carcinogenic potential of nanomaterials are evident (as shown in Figure 1). Following a comprehensive literature search, we compiled information from the selected studies and organized it in Table 1, which includes the name of the nanomaterials (particles), along with their physical-chemical characteristics, cell lines used, dose/concentration and duration of exposure, assays performed, significant findings, and references. To avoid redundancy, multiple studies from the same research group with the same transformation procedure were mentioned only once, while single studies with various NMs were organized separately in different rows.

### 5.1. Carcinogenic-Hallmark-Related Assays

Anchorage-independent growth (soft-agar colony formation assay) and migration and invasion assays were considered basic carcinogenic hallmark assays for almost all instances of NM-induced cell transformation when a human cell line was targeted. As exceptions, few studies chose only proliferation, migration, and invasion assays as proof of induced cell transformation [65,70,72]. Matrix metalloproteinases (MMPs), specifically MMP-2 and MMP-9, were used as biomarkers for NMNM-induced neoplastic transformation [37,39,42,43,60,72]. Several studies also performed wound-healing assays, angiogenesis, and xenograft tumorigenesis (in nude mice) (Table 1) as evidence of NM-induced in vitro transformation. A special emphasis was given to the ‘assays’ performed, which not only confirmed the cancer phenotypic hallmarks in transformed cells but are also linked with other mechanistic markers to delineate the malignant transformation process, such as oxidative stress, inflammation, DNA damage/repair (genotoxicity), epigenetic modification, apoptosis resistance, global (transcriptomics or proteomics) or selected gene/protein expressions, etc.

### 5.2. Cell Lines Applied

Human bronchial epithelial cells (BEAS-2B, 16HBE, HBEC-3KT), human small airway epithelial cells (SAEC), human pleural mesothelial (MeT5A), mouse embryonic fibroblasts (MEF), mouse colon epithelial cells (IMEC), mouse embryonic (NIH 3T3), and normal rat mesothelial (NRM2) cells were mainly applied in NM-induced cell transformation assays (Table 1). Besides those cell lines, Bhas 42 cell lines were used to perform OECD-test guideline-based CTAs (Table 1). In particular, a co-culture cell model (macrophages-like THP-1 and mesothelial, MeT5A) was used for induced cell transformation by MWCNT [70]. Some studies also include cancer cell lines from the same tissue as a positive control counterpart for specific biomarker analysis. For instance, transformed BEAS-2B cells were compared with A549 cells (pneumocyte type I-like cell line derived from a lung cancer patient) [51], while transformed normal rat mesothelial (NRM2) cells were compared with rat mesothelioma (ME1) for OPN expressions [65].

### 5.3. NM Exposure to Induce Cell Transformation

#### 5.3.1. Time of Exposure

The time required to induce cell transformation by nanomaterials ranges from 4 weeks to 26 weeks (Table 1). Most studies, with few exceptions, did not demonstrate a link between the cell carcinogenic hallmark with the time of exposure, so it is difficult to discuss the optimum exposure time point needed for malignant transformation for particular nanomaterials to indicated cell lines, as it depends on the choice of NM and the in vitro cell model combination. Dynamic changes during the exposure stage were analyzed with related assays (ROS, cell apoptosis, cell cycle, as well as MMP expressions, etc.) at different exposure stages (0, 3, 7, 15, 30, and 60 days); nonetheless, the carcinogenic phenotypic assays (cell migration, invasion, wound healing, soft-agar colony formation, in vivo xenograft) were only performed at the end of the exposure (60 days) and recovery (30 or 60 days) phase [74]. Two time points (6th week and 10th week) were chosen for the assessment of the carcinogenic potentiality of cerium oxide (CeO_2_-NP) and ferric oxide (Fe_2_O_3_-NP) NPs in SEAC. Clear time dependency was observed in Fe_2_O_3_-NP exposure with the cancer hallmark phenotypes, such as increased proliferation, invasion, and the ability to form colonies on soft agar [59]. In a similar pattern, the 8th and 12th weeks of exposure were selected for the cell transformation ability of pristine and functionalized MWCNTs. Nonetheless, time dependency was not clear in assessing cancer phenotypes among MWCNTs [69]. Time-dependent neoplastic phenomena were evident in AgNP exposed Caco-2 cells [43], Co-NP [37] and ZnO-NP [39] exposed MEF cells, and MWCNT (NM-400) and TiO_2_-NP materials (NM62002) treated human bronchial epithelial cell line (HBEC-3KT) [57]. Kornberg et al. demonstrated oscillatory time effects on anchorage-independent colony formation on BEAS-2B cells exposed to Fe_2_O_3_-NP (uncoated and silica coated), gas metal arc mild steel-welding fumes (GMAMS), and TiO_2_-NP [50]. Fe_2_O_3_-NP and GMAMS-exposed cells exhibited a significant increase (1.2- and 1.4-fold) in colony number at 83 days, gradually decreased by 111 days, and then became significantly elevated again (1.6-fold) at 138 days, which remained the same throughout the exposure period. In the case of TiO_2_-NP exposure, a significant increase (1.3-fold) in colony formation occurred between 111 to 138 days, which then ablated and returned to be similar to the control cells by 174 days and remained at baseline levels until the end of the exposure period.

#### 5.3.2. Exposure Concentrations

Mostly, a single sub-cytotoxic concentration was used to assess the NMs’ ability to induce cellular transformation; however, some studies also demonstrated a concentration-related carcinogenic potential by exposing NMs to various concentrations (Table 1). A concentration-dependent increase in cell transformation ability was demonstrated in Balb/c 3T3 [41] and Caco-2 cells [43], as well as induced malignancies in BEAS-2B cells exposed to Ag-NP [42]. Synthetic amorphous silica nanoparticle (SAS) (NM-200, NM-201, NM-202, and NM-203)-exposed Bhas 42 cell transformation model showed concentration dependency [45,46]. Only the highest concentration (20 µg/mL) showed significant cell transformation ability in TiO_2_-NP (NM102)-exposed BEAS-2B cells [54]. Cell transformation efficiency was found to be concentration-dependent in pristine and citrate-coated zirconium oxide (ZrO_2_-NP) and citrate-coated TiO_2_-NP [58] while not markedly concentration-dependent in Balb/3T3 cells exposed to MWCNTs [64]. In the same way, malignant transformation of BEAS-2B cells by MWCNT was not clearly concentration-dependent (i.e., significant soft-agar colonies were evident in the lowest (1 µg/mL) and the highest (20 µg/mL) but not in the medium concentration (10 µg/mL)) [55]. On the contrary, SWCNT exposure to MeT5A cells exhibited clear concentration dependency [73].

#### 5.3.3. Co-Exposure with Other Environmental Pollutants

Some studies have also taken into account the combined effects of exposure to other environmental pollutants. For instance, cigarette smoke condensate (CSC) has been evaluated as a secondary environmental pollutant that is often co-exposed with NMs [60,61]. In another study, the carcinogenic effects of polystyrene nanoplastics were evaluated in combination with Arsenic (As^III^), given that both agents are persistent water contaminants [63].

### 5.4. The Influence of Physicochemical Properties of NMs in Cell Transformation

Carbon-based nanomaterials (CNT), MWCNT and SWCNT (functionalized, aged, pristine), followed by silica nanoparticles are among the most studied NMs for cell transformation. Mainly CNT-related studies have demonstrated the connection between nano-specific (physicochemical) characteristics, such as surface functionalization, coatings, shape, size, and carcinogenic potentiality (induced cell transformation ability). The surface functionalization of CNTs (-NH_2_, -COOH, -OH, etc.) possesses a great impact on cell transformation efficiency, which may further differ with exposure concentrations. The –NH_2_-functionalized MWCNT showed significant anchorage-independent growth ability in soft agar but minimal ability of invasion in comparison to pristine and –COOH MWCNTs [64,69]. The increased cell transformation efficiency observed in the –NH_2_-functionalized MW-NHx may be attributed to its heightened reactivity with biomolecules, which could be a result of the reduced oxygen content and greater exposure of the carbon surface [69]. Shape as the determinant characteristic of MWCNT carcinogenic potentiality was documented when normal rat mesothelial (NRM2) cells were exposed to MWCNT, tangled vs. rigid. Exposure to the rigid, but not to the tangled, resulted in the transformation of NRM2 cells into an invasive phenotype. Furthermore, the study also postulated osteopontin (OPN) as a biomarker of in vitro cell transformation [65]. In general, the molecular mechanism of cell transformation, distinct from asbestos, shared similarities among CNTs (MWCNT vs. SWCNT). The studies with various carbon-based nanomaterials exhibited similar responses to malignant transformation hallmarks, such as migration and invasion [72], gene expression signatures [71], and cancer stem cell-like properties [67]. Size-related studies (diameter or aspect ratio) in CNTs did not obtain a clear outcome because the results variation is possibly related to the cell line of choice. MWCNT with a lower average diameter (NM-400) showed higher carcinogenic potentiality than that of a thicker diameter and high aspect ratio (NM-401), attested by soft-agar colony formation in the cell line HBEC-3KT (human bronchial epithelial cell) [57]. Conversely, mitsui-7, an MWCNT that possesses a morphological resemblance with NM-401, caused malignant transformation on human lung small airway epithelial cells (SAECs) [71].

Size-dependent cell transformation was observed when the HBEC-3KT cell line was exposed to the TiO_2_-NP and MWCNT with a smaller diameter but not to that with a larger diameter, in the soft-agar colony formation assay [57]. SAS (NM-200 and NM-201, NM-202 and NM-203) were reported as tumor-promoter substances in the Bhas 42 cell transformation model, and a marked size-dependency was apparent, in which a higher the particle size indicated a higher transformation ability [45]. In a comparison between coated (either with citrate or silica) vs. uncoated TiO_2_-NP and ZrO_2_NP, it was shown that coating with silica seems to prevent Balb/3T3 morphological transformation induced by ZrO_2_ NP [58]. In a similar line of evidence, silica-coated iron oxide nanoparticles (nFe2O_3_) do not induce neoplastic transformation in BEAS-2B cells [50]. A single study with nanocellulose (CNC) showed type/form (gel vs. powder)-specific carcinogenic potentiality assessed by anchorage-independent growth (of soft-agar colony formation), as well as cell migration assays [62].

Nanoceria (CeO_2_-NP) [59,60], nickel (NiNP) and nickel oxide (NiO-NP) nanoparticles [52] alone did not show clear cellular transformation, which was attested with in vitro cancer hallmark assays, such as soft-agar colony formation, cell migration, and invasion.

Contradictory results are reported for ZnO-NP, which did not induce cellular transformation in mouse embryonic fibroblast (MEF) [38,39] but was able to cause malignant transformation in mouse colon epithelial cells (IMECs) [40] and human embryonic kidney (HEK293) and mouse embryonic fibroblast [53] (NIH/3T3) cells [78], possibly through the CXCR2/NF-kB/STAT3/ERK and AKT pathways. An exact comparison between studies is difficult as these studies used different ZnO-NP with no similarity in physicochemical properties and no standardized protocol followed either. In a similar way of contradiction, while most studies reported positive cell transformation, some studies demonstrated no cell transformation in TiO_2_-NP [58] and SiO2-NP-exposed [49,50] cells.

### 5.5. Mechanism of NMNM-Induced Cell Transformation

#### 5.5.1. Oxidative Stress and Inflammatory Biomarkers

Oxidative stress, one of the main characteristics of carcinogens, is known to play a paradoxical role in cancer development. It can either support the transformation/proliferation of cancer cells by initiating/stimulating tumorigenesis or induce cell death [7,79]. Oxidative stress (such as reactive oxygen species (ROS) formation) associated with NM-induced neoplastic transformation was observed in various studies [37,39,50,53,54,55,59,62,74]. Nevertheless, none of the studies performed absolute phenotypic anchoring of carcinogenesis in connection with oxidative stress. For instance, whether low levels of ROS could alleviate NM-induced cell transformation capability by applying inhibitor/scavenger of ROS while NM treatment was not determined. Chronic inflammation predisposes to carcinogenesis at all stages of tumor formation [7,79]. BEAS-2B cells treated with cellulose nanocrystals (CNCs) (powder and gel form) cause the secretion of various pro- and anti-inflammatory cytokines, chemokines, and growth factors (IL-1β, IL-2, IL-4, IL-9, eotaxin, IL-1RA, IL-6, IL-8, G-CSF, IP-10, IL-15, TNF-α, PDGF-bb, RANTES, etc.), leading to neoplastic transformation [62]. A significant reduction in cytokine (IL-1β, IL-6, and IL-8) expressions was observed in correlation with MWCNT (NM403)-induced malignant transformation [55]. One study explicitly postulated the carcinogenicity of MWCNT through inflammation. The results from their long-term MWCNT-treated co-culture cell model (macrophages and mesothelial cells) indicated that IL-1β, secreted by macrophages, may significantly enhance the release of inflammatory cytokines (IL-8, TNF-α, and IL-6) from mesothelial cells. In particular, the NF-κB/IL-6/STAT3 pathway played a pivotal role in the malignant transformation of mesothelial (MeT5A) cells induced by MWCNTs [70].

#### 5.5.2. Genotoxicity, DNA Damage and Repair

Genotoxicity is among the most common and important characteristics of carcinogens [7]. It can be assessed by evaluating DNA damage, defects in the mechanisms of DNA damage repair, DNA damage response through several well-established in vitro assays, such as Ames, micronucleus and HPRT forward mutation assays (OECD genotoxicity test battery), comet assays, chromosomal aberration, altered expressions of DNA damage response/repair proteins, etc. These assays are continuously modified and adapted for safety assessments of nanomaterials [80,81]. The approaches adapted for genotoxicity evaluation in connection with carcinogenesis assessments are mainly the comet assay [37,39,52,54,55,58,62], the micronucleus assay [42,52,53,54,55,58,64,68], the chromosomal aberration assay [70], and the expressions of DNA damage response proteins (γ-H2AX and p53) [50,67]. The possibility of NM-induced oxidative DNA damage assessment cannot be ignored as NM-induced ROS formation and oxidative stress is a well-established phenomenon. DNA damage induced by oxidative stress and involvement of OGG1 gene, as well as the glycosylase of the base excision repair pathway which eliminates 8-oxoguanine lesions from DNA, were documented for cobalt (Co-NPs) and zinc oxide (ZnO-NP) nanoparticles, using the wild-type (M.E.F. Ogg1^+/+^) and isogenic knockout (M.E.F. Ogg1^−/−^) mouse embryonic fibroblast [37,39]. It was also shown that MTH1 could serve as a candidate biomarker to unravel NM (Co-NPs and ZnO-NP)-induced potential genotoxic and carcinogenic effects [38]. MTH1 is an important GTPase that helps to avoid the incorporation of oxidized nucleotides from the reservoir to the DNA by effectively degrading them. Nonetheless, most studies focus on DNA strand break but not on the mechanisms of genotoxicity, particularly how DNA damage repair is affected by NM exposure and in turn induces carcinogenesis. Cell-cycle aberrations as another potential mechanism of carcinogenesis have been reported in TiO_2_-NP [53] and amorphous SiO_2_-NP-induced [48] in vitro malignant transformation.

#### 5.5.3. Epigenetic Modifications

Epigenetic alteration is one of the critical characteristics of chemical carcinogens [7,82]; nonetheless, only a few studies have explored epigenetic alteration in connection with neoplastic-like transformation. Changes in epigenetic markers can be assessed by detecting global or gene-specific DNA methylation levels, histone modification (methylation, acetylation, phosphorylation etc.), or non-coding RNA (microRNA, lncRNA) expression levels [82]. Whole-genome DNA methylation microarray profiling delineated alteration of DNA methylation in cancer-related pathways as a potential underlying mechanism of post-chronic single-walled carbon nanotubes (SWCNTs) exposure-induced irreversible malignant transformation [74]. The integration of methylome and transcriptome data revealed altered genes expression, similar to lung adenocarcinoma and lung squamous cell carcinoma, for instance, promoter hypomethylation and upregulation of transmembrane serine protease 9 (TMPRSS9), proviral integration Moloney 2 (PIM2) genes or promoter hypermethylation and downregulation of calcium/calmodulin-dependent protein kinase II inhibitor 1 (CAMK2N1), and integral membrane protein 2A (ITM2A). Chronic exposure of nano silicon dioxide (Nano-SiO_2_)-induced malignant cellular transformation was associated with global DNA (5mC) hypomethylation and reduced expressions and enzyme activities of DNMTs (DNMT1, DNMT3A, DNMT3B), as well as altered expressions of methyl-CpG binding proteins (MeCEP2, MBD2) in two types of human bronchial epithelial cells (16HBE and BEAS-2B cells) [51]. Moreover, the demethylation of NRF2 promoter activates the expression of NRF2, which plays a key role in Nano-SiO_2_-induced carcinogenesis. A comparative carcinogenic potentiality in light of epigenetic modification (global DNA methylation) was carried out in Bhas42 mouse cell lines exposed to amorphous silica nanoparticles (NM-203) and crystalline silica particles (Min-U-Sil) [47]. Altered DNMT (DNMT1, DNMT3A, DNMT3B) expressions and global DNA (5mC) hypomethylation were evident in Min-U-Sil-exposed cells, but not in NM-203-exposed cells. In a similar line of evidence, increased histone (H4 but not H3) acetylation and modulated expressions of HDACs (HDAC1, HDAC2, HDAC3, and HDAC4) was observed only in Min-U-Sil-exposed cells. Furthermore, the transcriptional activation of the c-myc gene, a biomarker of carcinogenicity, through the regulation of epigenetic marks on its promoter was evident in both nano-silica-treated cells. Significant promoter modulation was observed for acetylated histone H3 lysine 4 (H3K4Ac), trimethylated histone H3 lysine 4 (H3K4me3), acetylated histone H3 lysine 9 (H3K9Ac), and acetylated histone H3 lysine 27 (H3K27Ac); however, no changes were evident for 5-methylcytosine (5-mC).

microRNA (miRNA) profiling, miRNA-mimic transfection, and gene and protein expressions analysis show that miR221 plays a critical role in MWCNT-induced neoplastic transformed cells. The miR221-ANNEXIN A1 axis was involved in the regulation of cell migration of the transformed cells [66]. Nanoceria CeO_2_-NP exposure gave rise to cell transformation (invasion and tumorsphere induction) in association with the altered battery of miRNA expression [61]. A small set of five miRNAs (miR-23a, miR-25, miR-96, miR-210, and miR-502) were reported as biomarkers for NM-induced transformed cells, which were validated particularly for NMs (TiO_2_NP, MWCNT, Co-NP, ZnO-NP, and CeO_2_-NP) [56]. A recent bioinformatics-based study highlights lncRNAs (in particular four lncRNA, namely MEG3, ARHGAP5-AS1, LINC00174, and PVT1) and pseudogenes (specifically five pseudogenes, namely MT1JP, MT1L, RPL23AP64, ZNF826P, and TMEM198B) as candidate diagnostic biomarkers and drug targets for CNT-induced lung cancer [83].

#### 5.5.4. Other Mechanisms of CTA-Induced NM

Regarding the causal or associative role of biomarkers in NM-induced carcinogenesis, apoptosis resistance is considered one of the key characteristics of malignant transformation [84]. The role of p53 in the apoptosis-resistance process of SWCNT-induced neoplastic transformation of lung epithelial cells (BEAS-2B) was reported among the pioneering studies related to NM-induced in vitro carcinogenicity [77]. Several follow-up studies were carried out to elucidate the underlying mechanism of oncogenesis of the same SWCNT-transformed cell model. SWCNT-transformed cells did present an aggressive phenotype, including increased cell migration, invasion, anchorage-independent cell growth (in vitro), and tumor formation and metastasis (in vivo). It was observed that Slug, a key transcription factor that induces an epithelial–mesenchymal transition (EMT) was a central player in these mechanisms [85]. Overexpression of mesothelin (MSLN) in SWCNT-induced neoplastic cell model and the potential application of MSLN as a biomarker and therapeutic target for CNT-induced malignancies [86]. Global gene expression analysis of the same model delineated activation of the pAkt/p53/Bcl-2 signaling axis, Ras family proteins for cell-cycle control, Dsh-mediated Notch 1, and the downregulation of apoptotic genes BAX and Noxa [87]. Other global gene expression studies evidenced that CNT-induced neoplastic-like transformation in normal mesothelial cells (MeT5A) was associated with overexpression of cortactin and H-Ras-ERK1/2 signaling (in the case of SWCNT exposure) [73], activating the MMP-2 gene and its critical role in an invasive phenotypic trait (MWCNT and SWCNT exposures) [72]. Whole-genome microarray further demonstrated differential signaling pathways between CNT-induced vs. asbestos-induced malignant transformation in primary small airway epithelial cells (SAECs). Conversely, CNTs (MWCNT and SWCNT) shared similar signaling pathways as an underlying mechanism of in vitro cell transformation. For instance, CNT-induced neoplastic cells demonstrated altered cell death, proliferation, mobility, development signaling, inflammation-related signaling, lipid metabolic signaling, TNFR signaling, reduced immune response, and altered cancer-related canonical pathways [71]. Global gene expression analysis reveals that inactivated p53 and aberrant p53 signaling comprise major reasons for the amorphous silica nanoparticle (SiNP)-induced malignant transformation of BEAS-2B cells [48]. Other than the global gene expressions analysis, targeted gene/protein expressions or enzyme activities shed light on the underlying mechanism of NM-induced malignant transformation. The results from silver nanoparticle (Ag-NP)-induced transformed cells indicate cell migration/invasion and apoptotic resistance by complex regulation of MAPK kinase (p38, JNK, and ERK1/2) and p53 signaling pathways [42].

### 5.6. NM-Induced Cancer Stem Cells (CSCs)

Research evidence indicates that cancer stem cells or stem-like cells (CSCs) are a subpopulation driving tumor initiation, progression, and metastasis. CSCs share characteristics with normal stem cells (NSCs); however, they have malignant phenotypic traits. CSCs can be identified based on stem cell surface markers, self-renewal capacity, potency for differentiation, resistance to apoptosis, unlimited proliferation, colony formation, formation of nonadherent spheroids, expression of epithelial–mesenchymal transition (EMT)-related transcription factors, and matrix metalloproteinase (MMP) secretion, xenograft tumor formation, etc. Particularly, CSCs are resistant to chemotherapy and sustain tumor growth and relapse after therapy [76,88,89]. Various CNTs (SWCNT, MWCNT, and ultrafine carbon black) showed CSC-like properties acquired through long-term exposure, as indicated by 3D spheroid formation, apoptosis resistance, and CSC marker expression through SOX2 and SNAI1 signaling [67]. SWCNTs caused the induction of CSC-like irreversible transformation with aberrant stem cell markers (Nanog, SOX-2, SOX-17, and E-cadherin and stem cell surface markers CD24^low^ and CD133^high^) [75], and their mechanistic insights were reported, such as the role of SOX9 overexpression in CSC formation and tumor metastasis [76].

### 5.7. NM-Induced Epithelial–Mesenchymal Transition (EMT)

The EMT process refers to the process by which transformed epithelial cells acquire the abilities involved in cell migration, invasion, and eventual cancer metastasis [2,90]. EMT induction by NM exposure shed light on the carcinogenic potentiality and need for cancer hallmark assessment. BEAS-2B cells co-cultured with THP-1-derived macrophages exposed to SiNPs promote EMT via the AKT pathway by inducing the release of SDF-1α and TGF-α while combined with benzo[α]pyrene-7, 8-dihydrodiol-9, 10-epoxide [91,92]. TiO_2_-NP can induce the EMT process in colorectal cancer (SW480) cells via the TGF-β/MAPK and WNT pathways [90]. MWCNT induces EMT in BEAS-2B cells via TGF-β-mediated Akt/GSK-3β/SNAIL-1 signaling pathway after extended (96 h) incubation at sub-cytotoxic concentrations [93]. A novel mechanism of CNT-induced carcinogenesis through the induction of cancer-associated fibroblasts, a critical tumor microenvironment component that provides the necessary support for tumor growth, has recently been described [94]. Moreover, the results also suggest the potential efficacy of podoplanin as a mechanism-based biomarker for rapid screening of carcinogenicity of CNTs and related NMs for their safer design [94]. RNA-seq analysis of silver nanoparticles exposed to BEAS-2B cells (1 µg/mL for 6 weeks) revealed fibrosis and the ‘epithelial–mesenchymal transition’ (EMT) pathway as a pivotal altered mechanism. Subsequent experimental validation supported the toxicogenomic analysis with increased collagen deposition, anchorage-independent cell growth, as well as cadherin switching [44].

In brief, although the exact mechanisms remain to be elucidated, the studied NMs (Table 1) have been found to have carcinogenic effects through the induction of oxidative stress, inflammation, genetic instability, apoptosis resistance, and EMT-related gene expression.

## 6. Future Research Needs for Better Nanosafety

Assessing the carcinogenic potential of a chemical or substance requires the incorporation of data from multiple endpoints and quantifiable phenotypes that are linked to molecular alterations [18]. CTAs have provided a useful tool for identifying carcinogenic agents, including non-genotoxic ones. However, relying solely on this endpoint is considered premature [95]. Mechanistic studies are crucial for evaluating chemical carcinogenesis, particularly in the absence of human data. Unfortunately, there are currently no universally accepted criteria for systematically identifying or organizing mechanistic data for decision making in carcinogenic potentiality assessments of chemicals [7]. One potential solution is to use the ten key characteristics of carcinogens (KCs) identified by the International Agency for Research on Cancer (IARC) in 2012 as a basic platform for evaluating the carcinogenic potential of chemical agents, including engineered nanomaterials (NMs) [7]. The data on these KCs exhibited by chemical agents can provide independent evidence of carcinogenic potentiality and can support grouping the same agent as “strong”, “moderate”, or “low”. Therefore, the combination of CTA with the assessment of the IARC KCs in parallel can provide more confidence in risk assessments.

Despite the usefulness of CTAs, only a few studies have paid attention to the underlying mechanisms of NM-induced cell transformation processes (Table 1). Mechanistic data can provide relevance of in vitro morphological transformation to tumorigenesis in vivo. The inclusion of mechanistic information generated from toxicogenomic studies in carcinogenicity and genotoxicity testing has been highlighted [96,97]. Moreover, recent studies have suggested that incorporating epigenetic information into the assessment of carcinogenic potentiality may enhance our understanding of the underlying mechanisms involved in carcinogenesis induced by environmental chemicals, including NMs [82]. Epigenetic alterations, such as changes in DNA methylation, histone modifications, and microRNA expression, have been shown to play a crucial role in regulating gene expression and cellular differentiation; they have also been associated with the development of various diseases, including cancer [82]. Emerging evidence suggests that NM exposure can lead to epigenetic alterations, which may contribute to their carcinogenic potentiality. For example, exposure to TiO_2_ (NM102) has been shown to induce neoplastic transformation without any genotoxic effects, suggesting a possible role of altered epigenetic status leading to carcinogenesis [54]. The causal or associative role of epigenetic biomarkers in NM-induced carcinogenesis could provide more confidence in nanosafety assessments. Future studies are needed to address which epigenetic alterations are most informative for specific types of chemical exposure-induced (including NMs) damage or disease, as well as how distinct epigenetic alterations are associated with nanomaterial carcinogenesis [80,82]. Furthermore, cell transformation assays with mechanistic assessment could be useful for the classification or grouping of genotoxic and non-genotoxic carcinogens and the long-term effects of NMs. Overall, incorporating mechanistic data into the assessment of the carcinogenic potentiality of NMs can enhance our understanding of the underlying molecular mechanisms and support better risk assessments for human health.

Some of the other key limitations are discussed as follows:The physical-chemical properties of NMs, such as their size, shape, surface modification/coating, and surface charges, are known to influence their carcinogenic potential. However, few studies have thoroughly examined the relationship between these properties and the potential for carcinogenesis, which is essential for promoting nanosafety and adopting a safe-by-design approach. Therefore, future study designs should prioritize assessing the safety of specific nanoforms in regard to their potential for carcinogenesis and evaluate them based on the ten key characteristics (10 KCs) recommended by the IARC. This will help identify which NMs pose the greatest risk for carcinogenicity and enable the development of safer nanomaterials through targeted modifications.It is a well-known fact that the oncogenic drive varies across species. Therefore, in the assessment of nanomaterials’ risk to humans, the transformation of human cells should be given more weight and considered more relevant compared to other species [98,99].Current studies on NM-induced cell transformation are primarily focused on pre-neoplastic changes, such as anchorage-independent growth in soft agar. However, these studies lack the confirmation of true malignancy through the mouse xenograft model, which is considered the gold standard for carcinogenicity evaluation. Therefore, future studies should consider finding alternative assays to replace the mouse xenograft test as a final step in assessing the true malignancy of NM-induced transformed cells.In vitro models, although useful for studying cellular transformation, do not fully capture the complexity of cancer formation in vivo, such as the role of the immune system and the tumor microenvironment. To bridge this gap, CTA can be combined with advanced microphysiological systems, such as organ-on-a-chip models or immune-oncology models, to better simulate in vivo situations [62]. However, single cell line-based CTA still has utility in elucidating the mechanisms of cellular transformation, which can provide insights into the formation of cancer at both the cellular and organism levels.While various cell transformation assays have been employed for carcinogenicity assessments of nanomaterials, there is a lack of standardization and harmonization among assays. Therefore, future studies should aim to establish standardized protocols and methods for cell transformation assays to enable better comparison of results and reproducibility.

## 7. Conclusions

This comprehensive review provides an overview of the current state of knowledge on applying CTA in the context of NM-induced carcinogenesis, focusing on the underlying mechanisms of cellular transformation. The utilization of CTA as a valuable addition to the repertoire of in vitro test systems for investigating the carcinogenicity of nanomaterials is supported by multiple lines of evidence.

Firstly, an appreciable number of studies has demonstrated the ability of CTA to identify information gaps and provide a point of comparison and reference for investigating the carcinogenic potential of nanomaterials. Secondly, the review suggests potential new niches for future studies using CTA. In addition, this review underscores the significance of employing CTA in conjunction with other tools and data, such as integrating with genotoxicity or toxicogenomics assays, for a comprehensive evaluation of the carcinogenicity of nanomaterials. Such a multi-disciplinary approach can augment the reliability and relevance of the findings and facilitate a more precise assessment of the carcinogenicity of nanomaterials. Lastly, as research on the carcinogenic effects of nanomaterials progresses, additional evidence is expected to enhance our understanding of the underlying mechanisms. This improved understanding can help develop effective strategies to mitigate potential risks to human health, such as establishing guidelines for safely handling nanomaterials, designing safer nanomaterials, and implementing appropriate risk assessment and management measures in workplaces.

In summary, the use of CTA in investigating nanomaterial-induced carcinogenesis is supported by evidence highlighting its ability to provide insights into the mechanisms underlying cellular transformation. When combined with other resources, CTA can be a valuable addition to the battery of in vitro test systems for assessing the carcinogenicity of nanomaterials and mitigating potential risks to human health. Further research in this field is essential for advancing our understanding of the intricate mechanisms underlying nanomaterial-induced carcinogenesis and developing strategies to ensure the safe and responsible use of nanomaterials in diverse applications.

## Figures and Tables

**Figure 1 ijms-24-08219-f001:**
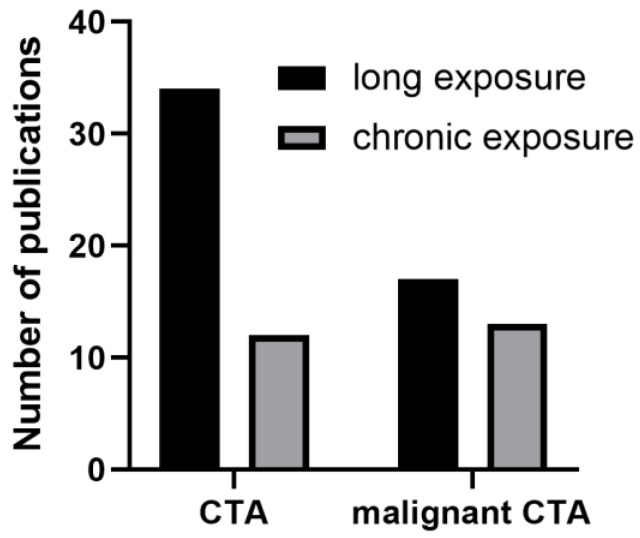
Cell transformation assays (CTAs) were performed with nanomaterials. A PubMed search for “long exposure of nanomaterials” or “chronic exposure of nanomaterials” followed by (malignant) cell transformation assay identified on the *X*-axis was performed in November 2022 (the figure is generated with the numbers of published papers appearing in the PubMed database with specific keywords). Of note, deleting the words “assay” yields even higher numbers for all categories.

**Table 1 ijms-24-08219-t001:** NM-induced Cellular Transformation (Publications until November 2022).

Nanomaterials (Name, Size, Surface Modification, Shape etc.)		Cell Model	Exposure		Assays				Main Findings	References
			Dose	Time	Cancer Phenotypic Hall Mark	Genotoxicity	Epigenetic Markers	Other Related Assays		
Cobalt Nanoparticles (CoNP)	Size (<50 nm), density (8.9 g/mL), surface area (>15 m^2^/g) *	Mouse embryonic fibroblasts (MEF Ogg1^+/+^) and MEF Ogg1^−/−^)	0.05 and 0.1 µg/mL	12 weeks	Anchorage-independent cell growth (soft-agar assay), morphology, MMP 2 & MMP 9 secretion,	Comet assay (DNA damage and oxidative DNA damage with FPG comet)	N/A	Cellular uptake, cell viability, ROS formation, gene expressions	CoNPs may pose a carcinogenic risk by inducing oxidative DNA damage, as suggested by increased sensitivity of MEF Ogg1^−/−^	[37]
	Size (<50 nm), density (8.9 g/mL), surface area (>15 m^2^/g) *	Mouse embryonic fibroblasts (MEF Ogg1^+/+^) and MEF Ogg1^−/−^)	0.1 µg/mL	12 weeks for MEF Ogg1^−/−^) & 6 weeks after MTH1 knockdown (KD) (shRNA)	Anchorage-independent cell growth (soft-agar assay), cell migration and invasion assays	N/A	N/A	Cell viability, MTH1 gene expressions	MTH1 (decreased phenotype in KD cell lines) is a significant contributor to NP-induced carcinogenicity	[38]
Zinc oxide Nanoparticles (ZnO-NP)	Size (<100 nm), surface area (>15–25 m^2^/g)	Mouse embryonic fibroblasts (MEF Ogg1^+/+)^ and MEF Ogg1^−/−^)	1 µg/mL	12 weeks	Anchorage-independent cell growth (soft-agar assay), cell migration and invasion assays	N/A	N/A	Cell viability, MTH1 gene expressions	MTH1 elicits as a relevant player in the NP-induced toxicity and carcinogenicity	[38]
	Size (<100 nm), surface area (>15–25 m^2^/g)	Mouse embryonic fibroblasts (MEF Ogg1^+/+^) and MEF Ogg1^−/−^)	1 μg/mL	12 weeks	Anchorage-independent cell growth (soft-agar assay), MMP2 and MMP9 secretion	Comet assay (DNA damage and oxidative DNA damage with FPG comet)	N/A	Cellular uptake, cell viability and internalization, ROS formation, gene expressions	Both cell types did not show any cellular transformation	[39]
	size distribution 35.6 ± 32.0 nm.	Mouse colon epithelial cells (IMEC)	1 μg/mL	30 passages	Anchorage-independent cell growth (soft-agar assay), wound-healing assay, xenograft tumorigenesis (in nude mice)	N/A	N/A	Cellular uptake, ROS formation, protein expression, knockdown of CXCR2	The CXCR2/NF-kB/STAT3/ERK and AKT pathways may be responsible for malignant transformation	[40]
Silver Nanoparticles (AgNP)	1–80 nm in size, diameter of <100 nm (average—80.0 ± 6.0 nm)	Balb/c 3T3 A31-1-1 mouse cell	0.17, 0.66, 2.65, 5.30, and 10.60 μg/mL	72 h	Cell transformation assay	A cytokinesis-block micronucleus (CBMN) assay	N/A	Cytoxicity assay (colony formation)	The frequency of morphological malignant transformation increased significantly in a dose-dependent manner	[41]
	1–80 nm in size, diameter of <100 nm (average—80.0 ± 6.0 nm)	BEAS-2B cells	0.13 and 1.33 µg/mL	4 months (40 passages)	Anchorage-independent cell growth (soft-agar assay), cell migration and invasion assays	N/A	N/A	Cell viability assay, EMT/MAPK proteins expressions, anti-apoptotic-related gene/protein expressions	The complex regulation of JNK, p38, p53, and ERK1/2 signalling pathways and activation of MMP-9/TIMP-1 were found to mediate malignant cell transformation	[42]
	8.52 ± 1.82 nm in size and 83.52 ± 0.70 nm in diameter	Caco-2 cells	0.5 and 1 µg/mL	6 weeks (assessment on 2nd, 4th, and 6th week)	Anchorage-independent cell growth (soft-agar assay), cell migration and invasion assays, secretion MMP2 and MMP9, ability to promote the growth of another tumor cell line (HCT116) with conditioned medium from 72 h exposed Caco-2 cells	N/A	N/A	Cellular uptake, measurement of release of Ag ion	Potential carcinogenic risk associated with long-term exposure	[43]
	Citrate coated Silver Nanoparticles size: 10 nm and 75 nm	BEAS-2B cells	1 µg/mL or approx. 0.2 µg/cm^2^	6 weeks (assessment on 2rd, 4th, and 6th week	Anchorage-independent cell growth (soft-agar assay), cell migration and invasion assays, E- and N-cadherin expression (EMT assays), collagen analysis	Comet and micronucleus assays	Genome-wide DNA methylation analysis	Intracellular uptake, transcriptomics analysis (RNA-seq)	Induce fibrosis (pro-fibrotic), EMT, and cell transformation	[44]
Silica nanoparticles	Synthetic amorphous silica nanoparticles (SAS) NM-200 and NM-201, NM-202 and NM-203 *	Bhas 42 cells	2 μg/cm^2^ to 80 μg/cm^2^	21 days	Bhas 42 cell transformation assay (by following OECD guidelines)	N/A	N/A	N/A	SAS may act as tumor promoters	[45]
	Synthetic amorphous silica nanoparticles (SAS) NM-203	Bhas 42 cells	1 μg/cm^2^ to 40 μg/cm^2^	21 days	Bhas 42 cell transformation assay (by following OECD guidelines)	N/A	N/A	Cell proliferation, transcriptomics (microarray)	A 12-gene signature could potentially serve as an early “bio-marker” of cell transformation	[46]
	Amorphous silica nanoparticle (NM-203) and crystalline silica particle (Min-U-SilVR 5)	Bhas 42 cells	NM-203: 0 to 5 μg/cm^2^; Min-U-SilVR 5: 0 to 25 μg/cm^2^	21 days Cell pellets were collected on Day 6 (D6) for epigenetic modification analysis.	No phenotype of cell transformation assays	N/A	Global DNA methylation, global histone acetylation (ELISA) with DNMTs and HDACs protein expressions, gene specific epigenetic analysis for c-Myc promoter (ChIP-qPCR)	c-Myc expression	Min-U-SilVR 5 reduced global DNA methylation and increased expression of DNMT3a, DNMT3b, histone H4 acetylation, and HDAC protein levels. NM-203 treatment showed no changes in epigenetic modification. Modulated parameters at D6 were restored in transformed cells at D21	[47]
	Amorphous silica nanoparticles (SiNPs) Diameter: 57.66 ± 7.30 nm	Human lung epithelial (BEAS-2B) cells	5 µg/mL	18 weeks (40 passages)	Enhanced cellular proliferation (MTT), anchorage-independent cell growth (soft-agar assay), and increased cell migration (wound-healing assay), xenograft tumorigenesis (in nude mice)	N/A	N/A	Morphology and proliferation assay, Cell-cycle assessment, genome-wide transcriptional analysis (microarray), gene and protein expressions (qRT-PCR and Western blotting)	Induced malignant transformation via p-53 signalling	[48]
	Amorphous silica nanoparticles (aSiO_2_NPs) NM-200, NM-203, NRT-808, NRT-817, NRT-820, NRT-944	Balb/3T3 mouse fibroblasts	1, 10, and 100 µg/mL	72 h	Balb/3T3 cell transformation assay, colony forming efficiency	Cytokinesis-block micronucleus assay	N/A	Cell viability	No cyto-genotoxic effect and no induction of morphological transformation	[49]
	Amorphous silica nanoparticles (SiO_2_-NP) size: 19 nm, surface area: 147 m^2^/g	Human lung epithelial (BEAS-2B) cells	~0.24 μg/cm^2^ delivered dose (0.6 μg/cm^2^ administered dose and)	6.5 months	Attachment-independent colony formation (soft-agar colony formation assay) (55, 83, 111, 138, 174, 202 days measurements)	Induction of double-stranded DNA damage (γ-H2AX immunostaining assay)	N/A	Particle uptake (TEM), cell proliferation (WST-1) and ROS production, intracellular iron and lysosome counts (LysoTracker)	No significant changes in attachment-independent colony formation throughout the exposure period	[50]
	Nano silicon dioxide (Nano-SiO_2_)	Human lung epithelial (16HBE and BEAS-2B cells and)	10.0 µg/mL for 16HBE cells and 40.0 mg/mL for BEAS-2B cells	32 passages for 16HBE and 45 passages for BEAS-2B cells	Anchorage-independent cell growth (soft-agar assay), wound-healing assay, enhanced cellular proliferation (MTT), xenograft tumorigenesis (in nude mice)	N/A	5mC content detection, DNMT enzyme activity, promoter methylation analysis (for NRF2 with MSP-PCR	Selected gene and protein expressions, cell transfection for NRF2 gene knockdown and overexpression	Induces malignant cellular transformation through global DNA hypomethylation. Demethylation of the NRF2 promoter activates NRF2 expression, which is essential in protecting against carcinogenesis induced by Nano-SiO_2_	[51]
Nickel nanoparticles (NiNPs)	Size: <100 nm, BET-area of 6.41 m^2^/g), nickel(II) oxide NPs (NiO, <50 nm, BET-area of 102 m^2^/g)	Human lung epithelial (BEAS-2B) cells	0.5 µg/mL	6 weeks	Anchorage-independent cell growth (soft-agar assay), cell migration and invasion assays	DNA strand breaks (comet assay), micronucleus (Flow Cytometric), cell cycle	N/A	Cytotoxicity, whole-genome gene expression analysis (RNA-seq), intracellular Ni level	No significant changes were observed in cell transformation or cell motility. DNA strand breaks were observed, but no induction of micronuclei was seen. Gene expression changes included calcium-binding proteins (S100A14 and S100A2) and genes such as TIMP3, CCND2, EPCAM, IL4R, and DDIT4	[52]
	Size: 20 nm, composed of anatase (90%) and rutile (10%), specific surface area is 43.8 m^2^/g	Human lung epithelial (BEAS-2B) cells	0.25 and 0.5 µg/mL	~150 days (21 cycles)	Anchorage-independent cell growth (soft-agar assay)	DNA damage response (DDR)-associated proteins expression	miRNA (miR-210) expression (qPCR)	HIF-1α/miR-210/Rad52 pathway gene expression	Exposure-induced DNA damage and DNA repair defects through HIF-1α/miR-210/Rad52 pathway likely contribute to genomic instability and ultimately cell transformation	[52]
TiO_2_ nanoparticles (TiO_2_-NP)	diameter <100 nm	Mouse embryonic (NIH 3T3)	10 μg/mL	12 weeks	Anchorage-independent growth assay (soft-agar colony formation assay), enhanced cellular proliferation (MTT and colony formation),	Micronucleus formation, cell-cycle analysis (flow cytometry), perturbed mitosis and cytokinesis	N/A	Cell viability, ROS, apoptosis, intracellular TiO_2_ level, MEK/ERK signalling pathway	Disrupts cell-cycle progression, causing chromosomal instability and cell transformation. PLK1 has been identified as the target for nano-TiO_2_ in the regulation of mitotic progression	[53]
	NM102: 21.7 ± 0.6 nm	Human lung epithelial (BEAS-2B) cells	0 to 20 μg/mL	4 weeks	Anchorage-independent growth assay (soft-agar colony formation assay)	Comet and micronucleus (MN) assays	N/A	Cell uptake, ROS	No ROS formation or genotoxic effects were observed, but there was a significant increase in transformed cell colonies, indicating a potential carcinogenic risk associated with nano-TiO_2_ exposure, which does not involve a genotoxic mechanism	[54]
	NM-102 size: 20.1 ± 7.4 nm *	Human lung epithelial (BEAS-2B) cells	10 µg/mL equivalent to 1.34 µg/cm^2^	6 weeks (endpoints analyzed at 3rd week and 6th week)	Anchorage-independent growth assay (soft-agar colony formation assay) in previously reported studies [54,55]	N/A	miRNA expression with qPCR (selected 33)	Cell viability, uptake	A set of five miRNAs (miR-23a, miR-25, miR-96, miR-210, and miR-502) were identified as informative biomarkers of NM-induced transformed cells	[56]
	NM62002 and KC7000 size: NM62002 1026 ± 895 nm, KC7000 4422 ± 1644 nm (FP7- NANoREG project)	Human bronchial epithelial cell line (HBEC-3KT)	1.92 and 0.96 μg/cm^2^	6 months (26 weeks)	Anchorage-independent growth assay (soft-agar colony formation assay) at 4, 8, 12, 16, 20, 24, and 26 weeks expansion of colonies picked from soft agar	N/A	N/A	Cytotoxicity	Exposure to NM62002, but not KC70000, led to cell transformation at week 12. However, the potential for colony formation was significantly reduced from weeks 12 to 16	[57]
	Aeroxide TiO2 (80:20 anatase/rutile structure) size: 23 nm	Human lung epithelial (BEAS-2B) cells	~0.57 μg/cm^2^ delivered dose (0.6 μg/cm^2^ administered dose and)	6.5 months (55, 83, 111, 138, 174, 202 days measurements)	Anchorage-independent growth assay (soft-agar colony formation assay)	Induction of double-stranded DNA damage (γ-H2AX immunostaining assay)	N/A	Intracellular uptake, cell proliferation, ROS production, intracellular iron and lysosome counts	Colony formation showed a significant 1.3-fold increase at 111 and 138 days but returned to levels similar to the non-treated control cells by 174 days and remained at baseline levels for the rest of the exposure period	[50]
	pristine (uncoated), surface modification with citrate and/or silica *	Balb/3T3 mouse fibroblasts	0 to 40 μg/cm^2^	72 h (followed by 31–35 days continued culture in clean media)	Balb/3T3 cell transformation assay, colony forming efficiency	DNA damage (comet assay), Cytokinesis-block micronucleus cytome assay	N/A	Cell viability, cell death (apoptosis/necrosis)	No cell transformation was evident	[58]
Zirconia nanoparticle (ZrO_2_-NP)	pristine (uncoated), surface modification with citrate and/or silica *	Balb/3T3 mouse fibroblasts	0 to 40 μg/cm^2^	72 h (followed by 31–35 days continued culture in clean media)	Balb/3T3 cell transformation assay, colony forming efficiency	DNA damage (comet assay), Cytokinesis-block micronucleus cytome assay	N/A	Cell viability, cell death (apoptosis/necrosis)	Induce cell transformation, except silica coated one	[58]
Iron Oxide Nanoparticle (Fe_2_O_3_-NP)	(i) no coating (nFe_2_O_3_) size: 19.6 nm, surface area: 42 m^2^/g) (ii) amorphous silica coating (SiO_2_-nFe_2_O_3_) size: 21.3 nm surface area: 49 m^2^/g (iii) gas metal arc mild steel welding fumes (GMA-MS) size: 15–45 nm *	Human lung epithelial (BEAS-2B) cells	nFe_2_O_3_ ~0.58 μg/cm^2^, SiO_2_-nFe_2_O_3_ ~0.55 μg/cm^2^ (delivered dose) (0.6 μg/cm^2^ administered dose)	6.5 months	Anchorage-independent growth assay (soft-agar colony formation assay)	Induction of double-stranded DNA damage (γ-H2AX immunostaining assay)	N/A	Intracellular uptake, cell proliferation, ROS production, intracellular iron and lysosome Counts	nFe_2_O_3_, but not SiO_2_-nFe_2_O_3_, induced a neoplastic-like phenotype, as evidenced by a significant increase in colony formation at 83 days and 138 days, which was maintained through the exposure period. No significant colony formation was observed at 55 days. GMA-MS-exposed cells had a significant increase in colony number	[50]
	ferric oxide (nFe_2_O_3_) nanoparticles	Human primary small airway epithelial cells (pSAECs)	0.6 μg/cm^2^	10 weeks (detection at 6th and 10th week)	Anchorage-independent cell growth (soft-agar assay), cell migration and invasion assays, increased proiferation	N/A	N/A	Intracellular uptake, ROS formation, CD71, DMT1, SLC40A1, FTH1expressions,	nFe_2_O_3_-exposed cells exhibited immortalization and retention of the malignant phenotype	[59]
Cerium oxide	cerium oxide (nCeO_2_) nanoparticles	Human primary small airway epithelial cells (pSAECs)	0.6 μg/cm^2^	10 weeks (detection at 6th and 10th week)	Anchorage-independent cell growth (soft-agar assay), cell migration and invasion assays, cell proliferation	N/A	N/A	Intracellular uptake, ROS formation, CD71, DMT1, SLC40A1, FTH1expressions	Increased proliferative capacity but no cell transformation ability	[59]
	size: 9.52 ± 0.66 nm with/without Cigarette smoke condensate (CSC)	Human lung epithelial (BEAS-2B) cells	1 and 5 μg/mL of CSC (CSC1 and CSC5), 2.5 μg/mL of CeO_2_NP alone or the Ce + CSC1 and Ce + CSC5	6 weeks	Anchorage-independent cell growth (soft-agar assay), cell proliferation, cell morphology, cell proliferation, wound-healing assay, secretion of MMP-9, FRA-1 as a biomarker of carcinogenesis	N/A	N/A	Cell viability, uptake (TEM), selected gene expressions	Although CeO_2_NP did not demonstrate any transforming ability, it was found to have a synergistic effect with CSC, enhancing the transforming effects of CSC and exacerbating the expression of FRA-1	[60]
	Size: <25 nm and density 7.13 g/mL) with/without Cigarette smoke	Human lung epithelial (BEAS-2B) cells	5 μg/mL of CSC, 2.5 μg/mL of nanoceria, and the combinations of both compounds (CeO_2_NPs plus CSC)	6 weeks	Invasion assay, tumorsphere formation assay	N/A	miRNA expression with qPCR (selected 33)	Cell viability, uptake (TEM)	Induces cell transformation and exhibits a positive interaction with the cell-transforming effects of cigarette smoke condensate	[61]
Nanocellulose	Cellulose nanocrystals (CNC) (powder and gel (10% wt.))	Human lung epithelial (BEAS-2B) cells	30 μg/cm^2^	4 weeks	Anchorage-independent cell growth (soft-agar assay), cell migration and invasion assays, cell proliferation, cell morphology	DNA damage (OxiSelect™ Comet assay)	N/A	Intracellular uptakes, oxidative stress assays, generation, inflammation marker assessment, apoptosis assay	Cellular transformation, enhanced invasion/migration, triggered oxidative stress and inflammatory response, and induced DNA damage were evident	[62]
Nano plastics	polystyrene nanoplastics (PSNPLs) size: 45.91 nm with/without Arsenic (AS^III^) *	Mouse embryonic fibroblasts (MEF Ogg1^+/+^) and MEF Ogg1^−/−^)	25 µg/mL PSNPLs; 2 µM As^III^, and combination of both (25 µg/mL PSNPLs + 2 µM As^III^)	12 weeks	Anchorage-independent cell growth (soft-agar assay), cell migration and invasion assays, cell proliferation, cell morphology, tumorsphere formation	DNA damage (comet assay)	N/A	Physical interaction of PSNPLs and AS^III^, intracellular uptake	Under co-exposed conditions, the PSNPLs showed the highest level of increased DNA damage and aggravated cellular transformation, followed by AS^III^. The general order of the tested endpoints was PSNPLs ≤ AS^III^ < co-exposure of (PSNPLs + AS^III^).	[63]
Carbon-based nanomaterials Multiwalled carbon nanotubes (MWCNTs) Carbon-based nanomaterials (CNMs) carbon nanotubes (CNTs) single-walled (SWCNT)	MWCNTs, diameter: 9.5 nm; length: <1–1.5 µm Pristine and Functionalised (-NH2, -OH, -COOH)	Balb/3T3	1, 10, and 100 µg/mL	72 h	Colony forming efficiency and cell morphological transformation (31 days)	Micronucleus assay	N/A	Cell uptake, cytotoxicity	Clear evidence of morphological transformation without cytotoxic and genotoxic effects	[64]
	tangled (tMWCNT) and rigid (rMWCNT)	Normal rat mesothelial (NRM2) cells	0.1 μg/mL	45 weeks (>85 passages)	Cell morphology, cell invasion	N/A	N/A	Osteopontin mRNA expressions (biomarker)	An invasive phenotype and increased OPN mRNA expression were observed in rMWCNTs, but not tMWCNT-exposed condition	[65]
	MWCNT diameter: 110 nm–170 nm; length: 5 μm−9 μm	Human pleural mesothelial (MeT-5A) cells	10 μg/cm^2^	1 year	Anchorage-independent growth (soft-agar assay), wound-healing assay	N/A	MicroRNA profiling	Application of miR221 mimics, ANNEXIN A1 expressions	The miR221-annexin a1 axis regulates cell migration in the induced transformed cells	[66]
	CNMs: MWCNTs SWCNTs UFCB ASB *	Primary human SAECs (immortalised with hTERT) cells	0.02 μg/cm^2^, equivalent to 0.1 μg/mL	6 months	Anchorage-independent growth (soft-agar assay), spheroid formation, Anoikis and apoptosis assays	DNA-strand breaks (γ-H2AX), DNA damage response (p53)		Stem cells marker assessment	Genotoxicity and CSC-like properties were evident in all CNM-exposed conditions. Gene signalling networks suggest involvement of SOX2 and SNAI1 signalling in cell transformation	[67]
	MWCNT (i) NM-400 diameter: 351 ± 140 nm and (ii) NM-401 Diameter: 710 ± 20 nm	human bronchial epithelial cell line (HBEC-3KT)	1.92 and 0.96 μg/cm^2^	6 months (26 weeks) (4th, 8th, 12th, 16th, 20th, 24th, and 26th week for assessments)	Anchorage-independent growth assay (soft-agar colony formation assay), expanding single colonies selected from soft agar	N/A	N/A	Cytotoxicity	NM-400, but not the agglomerated NM-401, showed cell transformation	[57]
	MWCNT, NM403 diameter: 12.0 ± 7.0 nm	Human lung epithelial (BEAS-2B) cells	1, 10 or 20 µg/mL	4 weeks	Anchorage-independent growth assay (soft-agar colony formation assay)	Comet assay and micronucleus (MN) assays	N/A	Detection of ROS and different interleukins (IL) such as IL-1B, IL-6 and IL-8, as well as HO-1	Increase in transformed cell colonies and decreased cytokine expression; no primary DNA damage but chromosome damage were observed	[55]
	MWCNTs Inner diameter: 2–10 nm Outer diameter: 10–30 nm Length: 1–30 µm	Human lung epithelial (BEAS-2B) cells	1 μg/mL (0.16 μg/cm^2^)	40 passages	Anchorage-independent growth assay (soft-agar colony formation assay), in vivo tumorigenicity assay	Cytokinesis-block micronucleus (CBMN) assay	N/A	Chromosomal instability (aCGH analysis), microarray, HOXD9 and HOXD13 gene function analysis (siRNA transfection)	Induction of irreversible oncogenic transformation and chromosomal aberration (in chromosome 2q31-32) may be attributed to HOXD9 and HOXD13, which are located in the same region	[68]
	Functionlised MWCNTs (fMWCNTs) (i) three-month aged as-prepared-(pMWCNT), (ii) carboxylated-(MW-COOH), and (iii) aminated-MWCNTs (MW-NHx) *	Human primary small airway epithelial cells (SAEC)	0.06 µg/cm^2^	12 weeks (8th and 12th weeks)	Anchorage-independent cell growth (soft-agar assay), cell migration and invasion assays, cell proliferation, cell morphology	N/A	N/A	Intracellular uptake	The surface properties of aged fMWCNTs can induce cell transformation, while exposure to pMWCNTs and MW-COOH also result in significant invasion behaviour	[69]
	MWCNTs diameter: 30–40 nm length: 10–20 µm	Mono as well as co-culturing macrophages (THP-1) and mesothelial (MeT5A) cells	0.1 mg/mL (in the co-cultured system, MeT5A cells in the upper chamber were exposed to MWCNTs only.)	3 months	Cell proliferation (every 24 h until 6 days), cell migration and invasion assay, colony formation assay (2 weeks)	Chromosome aberration assay	N/A	inflammatory cytokines (IL-1βIL-8, TNF-a, and IL-6) assay, NF-κB/IL-6/STAT3 pathway gene and protein expressions, transcriptomics	The NF-κB (p65)/IL-6/STAT3 pathway, induced by MWCNT-induced inflammation, played a crucial role in the malignant transformation	[70]
	MWCNT NM-401 size: 5.9 ± 4.6 nm) *	Human lung epithelial (BEAS-2B) cells	20 µg/mL of MWCNT, equivalent to 2.67 µg/cm^2^	6 weeks (endpoints analyzed at 3rd week and 6th week	Anchorage-independent growth assay (soft-agar colony formation assay) in previously reported studies [54,55]	N/A	miRNA expression with qPCR (selected 33)	Cell viability, uptake	A set of five miRNAs (miR-23a, miR-25, miR-96, miR-210, and miR-502) were identified as informative biomarkers of NM-induced transformed cells	[56]
	MWCNT Diameter: 81 ± 5 nm Length: 8.19 ± 1.7 µm and SWCNT Diameter: 1–4 nm Length: 1–4 µm *	Human primary small airway epithelial cells (SAEC)	0.02 μg/cm^2^ equivalent to 0.1 μg/mL	6 months	Anchorage-independent cell growth (soft-agar assay), cell migration and invasion assays, cell proliferation, cell morphology, angiogenesis assays	N/A	N/A	Intracellular uptake, whole-genome expressions (microarray)	MWCNTs and SWCNTs share similar gene signalling signatures that result in a neoplastic-like transformation phenotype	[71]
	MWCNT and SWCNT	Human pleural mesothelial (MeT5A)	0.02 μg/cm^2^ (sub cytotoxic)	4 months	Cell proliferation, cell migration and invasion, MMP-2 expressions	N/A	N/A	Whole-genome expression (microarray), expressions of MMP-2 and knockdown (shRNA)	Role of MMP-2 in CNT-induced cell transformation with cancer-like Properties, such as rapid growth and increased cell invasion and migration	[72]
	SWCNT	Human pleural mesothelial (MeT5A)	0.02, 0.06, and 0.2 μg/cm^2^	2 months	Anchorage-independent growth (soft-agar colony formation), cell invasion	N/A	N/A	H-Ras expressions and siRNA transfection, ERK1/2 expressions and inhibition	induced neoplastic transformation linked to H-Ras and ERK1/2 signaling	[73]
	SWCNTs outer diameter: <2 nm and length: 5–30 μm	Human lung epithelial (BEAS-2B) cells	10 μg/mL	60 days	Anchorage-independent cell growth (soft-agar assay), cell migration and invasion assays, wound-healing assay, in vivo tumorigenicity assay	N/A	Genome-wide DNA methylation arrays	Cell viability, ROS, cell apoptosis, cell cycle, MMP analysis	DNA methylation and transcriptome dysregulation, with enrichment in cancer-related pathways resulting in ‘irreversible’ transformation	[74]
	SWCNT Diameter: 0.8–1.2 nm Length: 0.1–1 μm *	Human small airway epithelial cells (SAECs)	0.02 μg/cm^2^ (physiologically relevant conc.)	6 months	Anchorage-independent cell growth (soft-agar assay), cell migration and invasion assays, apoptosis assay, tumor sphere assay, in vivo tumorigenicity assay	N/A	N/A	p53 (GFP) expressions, human stem cell proteome array, stem cell surface markers expressions	Irreversible malignant transformation and self-renewal, with in vivo tumorigenesis phenotypes and aberrant expression of stem cell markers (Nanog, SOX-2, SOX-17, and E-cadherin) and surface markers (CD24low and CD133high), indicating the presence of SWCNT-induced cancer stem cells	[75]
	SWCNT Diameter: 0.8–1.2 nm Length: 0.1–1 μm *	Human lung epithelial (BEAS-2B) cells	0.02 μg/cm^2^ equivalent to 0.1 μg/mL	6 months	Anchorage-independent cell growth (soft-agar assay), cell migration and invasion assays, tumor sphere assay, in vivo tumorigenicity assay	N/A	N/A	SOX9 expressions, knockdown, ALDH activity	SOX9 plays a role in the formation of SWCNT-induced cancer-stem-like cells, tumor metastasis, and the expression of stem cell marker ALDH1A1	[76]
	SWCNT Diameter: 0.8–1.2 nm Length: 0.1–1 μm	Human lung epithelial (BEAS-2B) cells	0.02 μg/cm^2^ equivalent to 0.1 μg/mL	6 months (24 weeks)	Anchorage-independent cell growth (soft-agar assay), cell migration and invasion assays, apoptosis assay, tumor sphere assay, angiogenesis assays, in vivo tumorigenicity assay	N/A	N/A	Protein array to evaluate apoptosis resistance mechanisms in transformed cells	p53-mediated apoptosis resistant in transformed cells	[77]

* Detailed physicochemical data available in paper.

## Data Availability

Not applicable.

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
