# Peer review of "In Vitro Cell Transformation Assays: A Valuable Approach for Carcinogenic Potentiality Assessment of Nanomaterials"

_ijms, 2023, doi:10.3390/ijms24098219_

Round 1
Reviewer 1 Report
Major comments:
This study is focused on "In vitro cell transformation assays –a valuable approach for carcinogenic potentiality assessment of engineered nanomaterials". The authors focused on describing method for assessing carcinogenic cell transformation assays and Engineered nanomaterials (ENM) and its Carcinogenicity assessments. The literature review doesn't clearly state the study objectives and their novelty. Research gaps are not clearly emphasized in the introduction part of the manuscript, so they should be more clearly written. The authors should highlight the importance of their review and its applicability in the future. Some paragraphs used in this manuscript are confusing and the sentences are not logically formed, which makes it harder to understand what the authors are trying to report in them. Abbreviations are simply too confusing. Also, the manuscript covers many different topics and some of them are not explained in detail. Therefore, there is no clear understanding of what the authors want to elaborate in terms of different parameters. There is little comparison between recent work and previous ones to highlight the importance of the work, the main problems of previously mentioned techniques are not emphasized. The scientific English language used in this manuscript needs major improvements. It is obvious the quality of the manuscript does not meet the standards of International Journal of Molecular Science in this form, therefore should need major revisions.
Specific comments:
1. The title includes a term engineered nanomaterials (ENM). As per my understanding engineered nanomaterials stands for modified/fabricated/hybrid materials. In this scenario, the term ENM does not match the discussion presented in this article. The data pretend in the Table 1: ENM induced Cellular Transformation includes data related to metal oxide nanopartilces, silica nanoparticles and Carbon nanomaterials.
2. Abstract doesn't highlight enough the novelty of this described approach. The exact applications of the study findings are also not emphasized enough which is insufficient to frame the whole picture of the present study.
2. The authors should address the goals of the study and emphasize research gaps, What should future studies in this area focus on? Please give some future work directions.
3. Please write the highlights or focus on that aspect more throughout the manuscript.
4. The discussion presented is very weak no strong comparison has been made with the literature to support the authenticity of the obtained results. Therefore, the authors are suggested to discuss their results with the following recent researches about types of nanomaterials, synthesis methods and novel materials to make the background and discussion more strong.
5. Introduction should be reconstructed by clearly explaining the problem statement i.e past, present and uniqueness of the work.
6. The discussion on ENM unique structures and properties should be expanded in the introduction.
7. It is better to list the typical routes of ENM that could cause cancer
8. It is suggested to add some representative figures from the publications into the manuscript for better illustrations and more direct understandings.
9. For a review paper, the main purpose is to summarize relevant published work and highlight the future development directions of the research field. However, the perspective/outlook part of this manuscript is very short and brief, which is not sufficient to give the readers constructive suggestions on how this area will be developed. The articles includes just one figure and one table
12. The conclusions only discuss a few of the researched criteria, which is insufficient to portray the full picture of this manuscript's contribution. The authors should offer detailed conclusions that include key values, and major discoveries, limitations of the study and recommendations for some future work.
The English of this manuscript should be polished. Long paragraphs are used in the manuscript, divide these long paragraphs into shorter paragraphs. Spelling correction required e.g. Line 86, insults
Author Response
Reviewer-1
Comments and Suggestions for Authors
Major comments:
This study is focused on "In vitro cell transformation assays –a valuable approach for carcinogenic potentiality assessment of engineered nanomaterials". The authors focused on describing method for assessing carcinogenic cell transformation assays and Engineered nanomaterials (ENM) and its Carcinogenicity assessments. The literature review doesn't clearly state the study objectives and their novelty. Research gaps are not clearly emphasized in the introduction part of the manuscript, so they should be more clearly written. The authors should highlight the importance of their review and its applicability in the future. Some paragraphs used in this manuscript are confusing and the sentences are not logically formed, which makes it harder to understand what the authors are trying to report in them. Abbreviations are simply too confusing. Also, the manuscript covers many different topics and some of them are not explained in detail. Therefore, there is no clear understanding of what the authors want to elaborate in terms of different parameters. There is little comparison between recent work and previous ones to highlight the importance of the work, the main problems of previously mentioned techniques are not emphasized. The scientific English language used in this manuscript needs major improvements. It is obvious the quality of the manuscript does not meet the standards of International Journal of Molecular Science in this form, therefore should need major revisions.
Answer: We appreciate reviewer’s comments and suggestion. To best of our knowledge, we tried to address the following points and carefully avoided the diverse parameters–
- What is carcinogenesis and carcinogenic agents and their properties
- What is CTA, types of CTA, its advantages and limitations
- The current state of the art of nanomaterials (NM)’s carcinogenicity evaluation
- Literatures discussion how CTA has been used for NM’s carcinogenicity evaluation
- Specifically from NM’s properties – size, shapes, surface functionalisation etc.
- Underlying Mechanism of NM induced carcinogenicity – oxidative stress, genotoxicity, epigenetic alterations and other mechanisms
- Point out ENM induced cancer stem cell
- Studies which demonstrated NM induced EMT process which is a one the pivotal step in invasion and migration
The aim of this manuscript was to discuss that CTA as a valuable platform for NM’s carcinogenicity which is in vitro method and satisfy 3R (replace, reduce, refine) principle. We have also mentioned where in other studies applied CTA. Specifically going into in-depth discussion NM vs other chemicals were beyond the scope of this review. There are specific reviews already avaible for metals. The principle carcinogenic assessment is still rodent-based 2 years study. Comparison with in vivo long term method was also the out of the scope of this review article.
Research gaps and future research need are discussed in details in the last section of the manuscript (section 6).
By following reviewer’s suggestion, we tried to homogeneous all the abbreviations throughout and in the end of the manuscript we added a list of abbreviation separately.
“The scientific English language used in this manuscript needs major improvements”
Answer: We worked on the language improvement through professional English editing system (please refer to the manuscript).
Specific comments:
- The title includes a term engineered nanomaterials (ENM). As per my understanding engineered nanomaterials stands for modified/fabricated/hybrid materials. In this scenario, the term ENM does not match the discussion presented in this article. The data pretend in the Table 1: ENM induced Cellular Transformation includes data related to metal oxide nanopartilces, silica nanoparticles and Carbon nanomaterials.
Answer: By following reviewer’s comment, we modified engineered nanomaterials (ENM) to nanomaterials (NM) all through the manuscript.
- Abstract doesn't highlight enough the novelty of this described approach. The exact applications of the study findings are also not emphasized enough which is insufficient to frame the whole picture of the present study.
Answer: We also provided a graphical abstract trying not to repeat the same information in the limited word number format in abstract.
- The authors should address the goals of the study and emphasize research gaps, What should future studies in this area focus on? Please give some future work directions.
Answer: Our manuscript has a full section dedicated to limitation and future direction that can be found in Section 6.
- Please write the highlights or focus on that aspect more throughout the manuscript.
Answer: We addressed all the points which were really relevant to the topic by avoiding unnecessary comparisons with other studies other than nanomaterials. Highlights are not asked as the format of the manuscript preparation.
- The discussion presented is very weak no strong comparison has been made with the literature to support the authenticity of the obtained results. Therefore, the authors are suggested to discuss their results with the following recent researches about types of nanomaterials, synthesis methods and novel materials to make the background and discussion more strong.
Answer: We did not understand the point of the reviewer here. This is not an original research paper addressing nanomaterials’ synthesis or characterization or toxicity results. No results presented in this manuscript. This is a literature review paper.
- Introduction should be reconstructed by clearly explaining the problem statement i.e past, present and uniqueness of the work.
Answer: In the ‘Introduction’ section, we provide the background of carcinogenesis to form the storyline to come to in vitro cell transformation assays. No previous review found in this topic related to CTA and nanomaterials, so no comparisons were necessary with previous work. Indeed all previous work tried to be documented in this review paper (until, November, 2022)
- The discussion on ENM unique structures and properties should be expanded in the introduction.
Answer: The main point is cell transformation assays induced by nanomaterials. We discussed in detailed about the state of the art of nanomaterials and their carcinogenicity assessment (Manuscript, Section 4, manuscript) and how the physico-chemical properties influence the CTA or their carcinogenic potentiality (Manuscript, Section 5.4). As this not a nanomaterials specific ‘original research article’, the detailed explanation about nanomaterials structure and properties is beyond the scope.
- It is better to list the typical routes of ENM that could cause cancer
Answer: The aim of this review paper is the ‘in vitro’ cell transformation assays. If we need to discuss in details about the typical routes of exposure for nanomaterials to cause cancer, we need to go through the in vivo studies which are beyond the scope of the present article. The general route of exposure of nanomaterials to cause any negative impacts on human including cancer has been mentioned (in the manuscript section 4).
- It is suggested to add some representative figures from the publications into the manuscript for better illustrations and more direct understandings.
Answer: Thanks for your comments. However, we did not find a suitable one. We provide a graphical abstract.
- For a review paper, the main purpose is to summarize relevant published work and highlight the future development directions of the research field. However, the perspective/outlook part of this manuscript is very short and brief, which is not sufficient to give the readers constructive suggestions on how this area will be developed. The articles includes just one figure and one table
Answer: To best of our knowledge, we gathered all the published work until November, 2022 related to CTA and nanomaterials. The aim was specific for nanomaterials and in vitro CTA. The number of published literatures are enough to organise in one table. In the table we made subsections for each type of nanomaterials.
The perspective and future research needs are discussed in detailed in the Section 6. We have discussed each limitation and their possible way out (Section 6) point by point. The future advancement will provide the scope to elaborate this more.
- The conclusions only discuss a few of the researched criteria, which is insufficient to portray the full picture of this manuscript's contribution. The authors should offer detailed conclusions that include key values, and major discoveries, limitations of the study and recommendations for some future work.
Answer: By following reviewer’s comments, we have modified the ‘conclusion’ part of the manuscript (please refer to the manuscript).
Comments on the Quality of English Language
The English of this manuscript should be polished. Long paragraphs are used in the manuscript, divide these long paragraphs into shorter paragraphs. Spelling correction required e.g. Line 86, insults
Answer: We worked on language improvement through the English editing system.

Reviewer 2 Report
In this manuscript the authors made a comprehensive review on carcinogenicity assessment of engineered nanoparticles by cell transformation assays (CTA). Different types of CTA were introduced and discussed by the authors. Examples of CTA to assess the carcinogenic potential of some popular nanoparticles in recent years were discussed by the authors. The manuscript may be of interesting for publication by taking some suggestions into consideration.
1) While the advantages and potentials of CTA were comprehensively discussed, should the limitations and drawbacks of CTA be discussed also?
2) Besides environmental pollutant exposure, stimulants exposure may be included and discussed, as many nanomaterials behave differently under different stimulants such as temperature, pH, light, electromagnetic waves etc.
3) Term “Engineered nanomaterials” seems too general, especially when only inorganic nanoparticles were included in the manuscript. Large numbers of nanomaterials are organic structures, such as polymers, artificial enzymes and nanopipetides. On the other hand, inorganic or not, nanoparticles are only one type of nanomaterials. Perhaps, a more specific or narrower term such as “inorganic nanoparticles” should be used.
4) Inconsistency of the font, font size and typography occurred many times throughout the paper. Such as Line 56 – 58, line 91 – 109, line 246 – 264 and line 652 – 687. Some typos, such as line 372.
There are some grammar issues and typos occurred throughout the manuscript. Proof reading is needed to solve the issues.
Author Response
Reviewer- 2
Comments and Suggestions for Authors
In this manuscript the authors made a comprehensive review on carcinogenicity assessment of engineered nanoparticles by cell transformation assays (CTA). Different types of CTA were introduced and discussed by the authors. Examples of CTA to assess the carcinogenic potential of some popular nanoparticles in recent years were discussed by the authors. The manuscript may be of interesting for publication by taking some suggestions into consideration.
Answer: We appreciate your positive comments and the answers for each questions are as follows -
1) While the advantages and potentials of CTA were comprehensively discussed, should the limitations and drawbacks of CTA be discussed also?
Answer: Thanks for your comment. The limitations of CTA van be found in the manuscript Section 3.2 (“CTA, based on the characteristics of transformed cell”, 3rd Paragraph).
2) Besides environmental pollutant exposure, stimulants exposure may be included and discussed, as many nanomaterials behave differently under different stimulants such as temperature, pH, light, electromagnetic waves etc.
Answer: We did not find any literatures conducted studies with nanomaterials with environmental factors (temperature, pH, light, electromagnetic waves etc.). We only found literatures as co-exposure conditions with cigarette smoke and arsenic and those has been included.
3) Term “Engineered nanomaterials” seems too general, especially when only inorganic nanoparticles were included in the manuscript. Large numbers of nanomaterials are organic structures, such as polymers, artificial enzymes, and nanopipetides. On the other hand, inorganic or not, nanoparticles are only one type of nanomaterials. Perhaps, a more specific or narrower term, such as “inorganic nanoparticles,” should be used.
Answer: We appreciate your comment! We have replaced the term ‘Engineered nanomaterials’ with ‘Nanomaterials’ in the manuscript. From our side, we were not specific to searching (or including) the literature of inorganic or organic nanomaterials/nanoparticles; rather was open to including all avaible literatures based on CTA and nanomaterials. We did not find organic nanomaterials related CTA studies, except one with polystyrene nanoplastics (PSNPLs) (section 5.3.4) (Ref. 60)
4) Inconsistency of the font, font size and typography occurred many times throughout the paper. Such as Line 56 – 58, line 91 – 109, line 246 – 264 and line 652 – 687. Some typos, such as line 372.
Answer: We have corrected those inconsistencies
Comments on the Quality of English Language
There are some grammar issues and typos occurred throughout the manuscript. Proof reading is needed to solve the issues.
Answer: We worked on language improvement through the English editing system (please refer to the manuscript).

Round 2
Reviewer 1 Report
All the comments have been adddressed by the authors
Author Response
Thanks for your comments.
Reviewer 2 Report
Authors have solved the issues in original manuscript. However, inconsistency of the font, font size or typography occurred in new manuscript (line 687 - 713). Please revise accordingly.
English is fine to read and understand.
Author Response
Thanks for your comments. The Editorial Office helps edit the layout of the revised version.